# Water quality assessment based on Nemerow pollution index method: A case study of Heilongtan reservoir in central Sichuan province, China

Kai Su[1]*, Qin Wang[1], Linxiao Li[1], Rong Cao[1], Yingwei Xi[2], Guizhi Li[3]

1 Faculty of Geosciences and Environmental Engineering, Southwest Jiaotong University, Chengdu, China, 2 Department of Pollution Source and Emergency Monitoring, Sichuan Ecological Environment Monitoring Station, Chengdu, China, 3 Department of Water Quality Monitoring, Sichuan Ecological Environment Monitoring Station, Chengdu, China

* ksu@swjtu.edu.cn

**Data Availability Statement:** All relevant data are within the manuscript.

**Funding:** This study is supported by Sichuan Science and Technology Program (2021YFS0284), The funders had no role in study design, data

## Abstract

In this paper, three monitoring sections were set up in Heilongtan Reservoir, and water samples were collected in 2019, 2020, and 2021 for the determination of physical and chemical properties such as permanganate index, chemical oxygen demand, and biochemical oxygen demand ($BOD_5$). The water quality was evaluated by the single factor pollution index method and the Nemerow pollution index method, and the temporal and spatial changes of water quality were analyzed. The single factor pollution index method determines the water quality category by identifying the single worst indicator of water quality, based on the classified water quality category. The Nemerow pollution index method emphasizes the most polluting factor while also taking into account the contribution of other factors in the assessment system, and determines the water quality category through the comprehensive pollution index. The results of the study indicate that the monitoring indicators of the monitoring sections have reached the Category III water quality standard and above in the "Surface Water Environmental Quality Standard" during the three years 2019 to 2021. The Heilongtan Reservoir's water quality in 2019, 2020, and 2021 is of Category I standard, according to the results of the evaluation of water quality using the single factor pollution index technique. According to the Nemerow pollution index method's results for evaluating water quality, the water quality pollution index for the three monitoring sections as a whole ranges from 0.36 to 0.51 in three years. The three monitoring sections' water quality—Dongfeng Canal, Longmiao, and Sixin Village—has not changed significantly during that time, remaining clean. In terms of temporal and spatial rates of change, the temporal rate of change (T) and spatial rate of change (S) over the three years were less than 20%, and the changes in water quality at each monitoring site were not significant.

collection and analysis, decision to publish, or
preparation of the manuscript.

**Competing interests:** The authors declared no
potential conflicts of interest with respect to the
research, authorship, and publication of this article.

## Introduction

In order to make full use of freshwater resources, people build reservoirs to meet the normal production and domestic water needs in the dry season. At the same time, a well-built reservoir is as important a part of the ecological environment as a lake, maintaining the ecological balance upstream and downstream of the reservoir, protecting the local ecological environment, regulating and maintaining the local climate [1,2]. It is crucial to stay on top of water quality changes in order to balance the use and protection of water resources in reservoirs. Therefore, by monitoring changes in the reservoir's water quality and making time-sensitive adjustments to reservoir consumption and protection measures, we may realize the goal of effective water resource utilization without causing ecological environment harm.

At present, the common methods used for water environment quality evaluation mainly include single factor evaluation method [3], comprehensive pollution index method [4], fuzzy comprehensive evaluation method [5], Nemerow pollution index method [6] and water quality index method [7]. The application of various water quality assessment methods in lakes and reservoirs has been relatively mature [8]. For example, Qinbang Sun et al. [9] used the single factor pollution index evaluation method and the comprehensive pollution evaluation method to evaluate the pollution status of heavy metals. Zhendong Hu et al. [10] proposed a new water quality comprehensive pollution index method to realize the evaluation of water quality. Ren Wenhao et al. [7] clarified the characteristics of water quality to provide a reference for the overall prevention and control of rivers in Zhengzhou, and used the comprehensive pollution index method and the Nemerow pollution index method to evaluate the water quality of 10 rivers in the main urban area of Zhengzhou. Based on the historical water quality data of Zhangze Reservoir in the past five years, Qian Zhang [11] used the comprehensive water quality identification index method and the Nemerow pollution index method to evaluate the water quality of Zhangze Reservoir.

The Hailongtan Reservoir is situated 16 kilometers northwest of Renshou County, near the western foot of the Er'e Mountain in the Longquan Mountains and the southeastern boundary of the Chengdu Plain. Additionally, the Hailongtan Reservoir, which provides drinking water for more than 3 million people in Renshou County and the neighboring counties and cities, is a protected natural green water source. The surrounding economy and tourism industry have grown quickly in recent years as a result of Heilongtan Reservoir's rapid development as a key tourism resource. However, at the same time, the reservoir's water quality has deteriorated to the point of eutrophication due to agricultural non-point source pollution and the discharge of domestic and production sewage [12]. Therefore, the environmental protection work of Heilongtan Reservoir is faced with greater pressure and challenges while meeting the needs of local development. From previous studies, we know that the single factor pollution index method and the Nemerow pollution index method are more mature and have been widely used in water quality assessment of other large and small lakes and reservoirs, providing a scientific basis for the protection of water resources. And yet, the related research on the water quality of Heilongtan Reservoir mostly focuses on the evaluation and analysis of water eutrophication. For example, Wang Meiqin et al. [13] evaluated the water eutrophication of Heilongtan Reservoir based on the comprehensive nutrient state index method. In the research on the water quality of Heilongtan Reservoir, there are few studies using the single factor evaluation method and the Nemerow pollution index method to evaluate it. However, the situation in Heilongtan Reservoir is similar to that of the lake reservoirs evaluated for water quality by the Nemerow pollution index method. Therefore, this paper uses the water quality monitoring data from 2019 to 2021 to evaluate and

analyze the water quality of Heilongtan Reservoir using the single factor evaluation method and the Nemerow pollution index method, so as to obtain the changes and good and bad water quality of the reservoir in three years, thus providing a scientific and theoretical basis for the environmental protection and sustainable development of Heilongtan Reservoir. The limitation of this study is that it cannot effectively predict the variation pattern of reservoir water quality and the influence of a single indicator factor on the evaluation results is large, which is expected to be overcome in future studies.

## Materials and method

### Research areas and data

Heilongtan Reservoir is located in the central and southern part of the Sichuan Basin, about 10 km away from Renshou County, at the southern foot of Longquan Mountain, 64 km away from Chengdu in the north, 83 km away from Leshan and Mount Emei in the south, and 30 km away from Meishan. The reservoir area is 186.4 km$^2$, 25 km long from north to south, 13 km wide from east to west, 310 km long by the lake, 23.6 km$^2$ on the lake surface, 23.6 km$^2$ in water area, and has a maximum water storage of 360 million m$^3$. A study showed that the annual average water quality of the Heilongtan Reservoir at the three points of Dongfeng Canal, Longmiao and Sixin Village over the five years from 2014 to 2018 was all surface water quality category III standard, with stable water quality [12–14]. In this paper, the water quality evaluation of Heilongtan Reservoir is carried out according to the three groups of data of Dongfeng Canal, Longmiao, Sixin Village and the whole.

### Research methods

**Collection and determination of water samples.**   Water samples were collected in 2019, 2020, and the first half of 2021 in accordance with the standards of the national surface water environmental quality monitoring network for collection, measurement and separation (on-site monitoring technical guidelines). Permanganate Index, COD, BOD$_5$, TP, Ammonia Nitrogen, Mercury, Lead, Cadmium, Chromium (hexavalent), Arsenic, Copper, Selenium, Fluoride, Cyanide, Sulfide, petroleum (petroleum ether extraction) and volatile phenol of water samples was carried out in the laboratory according to the national standard method.

**Water quality evaluation method.**   *(1) single factor pollution index method*. Single factor index method is in all involved in the comprehensive water quality assessment of water quality indicators, using the worst water quality single indicator belongs to the category to determine the water comprehensive water quality category [15], through the calculation of pollution index to determine the main pollutants in the water body and their degree of harm, single factor pollution index method of the formula is:

$$p_i = \frac{C_i}{S_i} \tag{1}$$

In the formula, Pi represents the pollution index of single water quality index i; Ci is the measured value of pollutant content (mg/L); S$_i$ is the standard value of environmental quality (mg/L), which is the Category III water standard in the Environmental Quality Standards for Surface Water (GB3838-2002).

The environmental quality standard evaluation grading standard of the single factor pollution index method is shown in Table 1.

*(2) Nemerow pollution index method*. Compared to the single factor index method, the Nemerow Pollution Index method is a more comprehensive approach to water quality

Table 1. Water quality category determination based on the single factor pollution index method.

| Water quality category | $P_i$ | Pollution assessment |
|---|---|---|
| I | $\leq 1$ | No pollution |
| II | (1,2] | Slightly pollution |
| III | (2,3] | Lightly pollution |
| IV | (3,5] | Moderately polluted |
| V | >5 | Seriously pollution |

assessment, which emphasizes the most polluting factors while also taking into account the contribution of other factors in the assessment system [16], and determines the water quality category by calculating a comprehensive pollution index, which is a weighted multi-factor environmental quality index that takes into account the extreme values or highlights the maximum values [17], and the comprehensive pollution index is calculated as follows:

$$P_N = \sqrt{\frac{(P_1)^2 + P_{imax}^{\;2}}{2}} \tag{2}$$

In the formula, $P_N$ is the comprehensive pollution index of the sampling point; $P_{imax}$ is the maximum value of the single-item pollution index of the pollutants at the sampling point; $P_1 = \frac{1}{n}\sum_{i=1}^{n} P_i$ is the average value of the single-factor index.

The grading standard for environmental quality evaluation by the Nemerow pollution index method [18] is shown in Table 2.

*(3) Spatial and temporal variation characteristics of water quality in Heilongtan Reservoir.* The degree of water quality change with time and space is judged according to the time change rate T and the space change rate S of water quality, respectively. The calculation formula of T and S are as follows [19],

$$T = \frac{G_{t1} - G_{t2}}{G_{t1}} \times 100\% \tag{3}$$

$$S = \frac{G_{s1} - G_{s2}}{G_{s1}} \times 100\% \tag{4}$$

In the formula, $G_{t1}$ is the Nemerow index at the start time of the comparison time. $G_{t2}$ is the Nemerow index of the sampling point at the termination time; $G_{S1}$ and $G_{S2}$ are the Nemerow index in the sampling points of the start section and the end section in the comparison space, respectively.

## Data processing

Excel software was used for data processing, and Origin 2021 software was used for graphing.

Table 2. Water quality level determination based on the Nemerow pollution index method.

| Water quality level | $P_N$ |
|---|---|
| I | <0.59 |
| II | [0.59,0.74) |
| III | [0.74,1.00) |
| IV | [1.00,3.50) |
| V | $\geq 3.50$ |

**Table 3. Statistical results of various indicators in Heilongtan Reservoir in 2019.**

| Monitoring indicators | Monitoring section | | | | | | | |
|---|---|---|---|---|---|---|---|---|
| | Dongfeng Canal | | Longmiao | | Sixin Village | | The whole | |
| | Average value | Measured category | Average value | Measured category | Average value | Measured category | Average value | Measured category |
| Permanganate Index | 1.79 | I | 2.63 | II | 2.57 | II | 2.33 | II |
| BOD$_5$ | 1.5 | I | 1.6 | I | 1.6 | I | 1.5 | I |
| Ammonia nitrogen | 0.143 | I | 0.112 | I | 0.114 | I | 0.123 | I |
| Petro | 0.0158 | I | 0.0079 | I | 0.0088 | I | 0.0108 | I |
| Volatile phenol | 0.00015 | I | 0.00015 | I | 0.00015 | I | 0.00015 | I |
| Hg | 0.00002 | I | 0.00002 | I | 0.00002 | I | 0.00002 | I |
| Pb | 0.005 | I | 0.005 | I | 0.005 | I | 0.005 | I |
| Cd | 0.0005 | I | 0.0005 | I | 0.0005 | I | 0.0005 | I |
| Anionic surfactant | 0.029 | I | 0.025 | I | 0.025 | I | 0.0263 | I |
| Cr(hexavalent) | 0.002 | I | 0.002 | I | 0.002 | I | 0.002 | I |
| Fluoride(F⁻) | 0.171 | I | 0.186 | I | 0.161 | I | 0.173 | I |
| TP(P) | 0.0308 | III | 0.0241 | II | 0.0254 | III | 0.0268 | III |
| Cyanide | 0.002 | I | 0.002 | I | 0.002 | I | 0.002 | I |
| Sulfide | 0.0027 | I | 0.0025 | I | 0.0025 | I | 0.0026 | I |
| As | 0.0010 | I | 0.0011 | I | 0.0010 | I | 0.0011 | I |
| CODcr | 7.25 | I | 9.08 | I | 9.00 | I | 8.44 | I |
| Cu | 0.025 | II | 0.025 | II | 0.025 | II | 0.025 | II |
| Zn | 0.025 | I | 0.025 | I | 0.025 | I | 0.025 | I |
| Se(Tetravalent) | 0.0002 | I | 0.0002 | I | 0.0002 | I | 0.0002 | I |

Note: The unit of each index is mg/L.

## Results and discussion

### Heilongtan Reservoir water quality index results

The statistical results of water quality indicators of Heilongtan Reservoir in 2019, 2020 and 2021 are shown in Tables 3–5 respectively. It is found that in 2019, total phosphorus was classified as Category III standard, copper as Category II standard, and the rest of the indicators all meet the Category I standard of "Surface Water Environmental Quality Standard" (GB3838-2002). In 2020, the permanganate index will be reduced from Category I to Category II, and the rest of the indicators meet the same categories as in 2019. The permanganate index, ammonia nitrogen, and total phosphorus will all meet the Category II standard in 2021, copper will be upgraded to the Category I standard, and the rest of the indicators will remain the Category I standard. According to the data comparison for three consecutive years, the overall water quality of Heilongtan Reservoir is excellent, and most of the monitoring indicators in the three years meet the Category I standard. In 2021, all indicators in the monitoring section will meet the Category I and II standards of the "Surface Water Environmental Quality Standard", which will improve the water quality compared to 2019.

### Evaluation of water quality by single factor pollution index method

According to the Formula (1), the data of the three monitoring indicators in the Heilongtan Reservoir are calculated, and the single factor pollution index value of each monitoring indicator in the Heilongtan Reservoir is shown in Table 6 below. The single factor index method is

**Table 4. Statistical results of various indicators in Heilongtan Reservoir in 2020.**

| Monitoring indicators | Monitoring section | | | | | | | |
|---|---|---|---|---|---|---|---|---|
| | Dongfeng Canal | | Longmiao | | Sixin Village | | The whole | |
| | Average value | Measured category | Average value | Measured category | Average value | Measured category | Average value | Measured category |
| Permanganate Index | 2.78 | II | 2.51 | II | 2.63 | II | 2.64 | II |
| BOD$_5$ | 1.6 | I | 1.7 | I | 1.7 | I | 1.7 | I |
| Ammonia nitrogen | 0.118 | I | 0.108 | I | 0.112 | I | 0.112 | I |
| Petro | 0.0075 | I | 0.005 | I | 0.006 | I | 0.006 | I |
| Volatile phenol | 0.00015 | I | 0.00015 | I | 0.00015 | I | 0.00015 | I |
| Hg | 0.00002 | I | 0.00002 | I | 0.00002 | I | 0.00002 | I |
| Pb | 0.005 | I | 0.005 | I | 0.005 | I | 0.005 | I |
| Cd | 0.0005 | I | 0.0005 | I | 0.0005 | I | 0.0005 | I |
| Anionic surfactant | 0.025 | I | 0.025 | I | 0.025 | I | 0.025 | I |
| Cr(hexavalent) | 0.002 | I | 0.002 | I | 0.002 | I | 0.002 | I |
| Fluoride(F$^-$) | 0.184 | I | 0.161 | I | 0.179 | I | 0.175 | I |
| TP(P) | 0.035 | III | 0.016 | II | 0.0175 | II | 0.0228 | II |
| Cyanide | 0.002 | I | 0.002 | I | 0.002 | I | 0.002 | I |
| Sulfide | 0.0025 | I | 0.0025 | I | 0.0025 | I | 0.0025 | I |
| As | 0.0012 | I | 0.0012 | I | 0.0013 | I | 0.0012 | I |
| CODcr | 10.00 | I | 10.50 | I | 9.67 | I | 10.06 | I |
| Cu | 0.025 | II | 0.025 | II | 0.025 | II | 0.025 | II |
| Zn | 0.025 | I | 0.025 | I | 0.025 | I | 0.025 | I |
| Se(Tetravalent) | 0.0002 | I | 0.0002 | I | 0.0002 | I | 0.0002 | I |

Note: The unit of each index is mg/L.

based on the three types of water quality standards in the Environmental Quality Standards for Surface Water, and the water quality evaluation results are shown in Fig 1 below. The water quality evaluation levels for the three years 2019, 2020, and 2021 for monitoring indicators like permanganate index and biochemical oxygen demand (BOD$_5$) are all Category I standards, and the water quality evaluation outcomes for the three years for Heilongtan Reservoir using the single-factor pollution index method are all Category I water quality.

Compared with the water quality standards in the Environmental Quality Standards for Surface Water, the single factor pollution evaluation method can more simply and intuitively reflect the pollution status of the water quality of Heilongtan Reservoir. The water quality of Heilongtan Reservoir is Category I standard, which is pollution-free.

## Evaluation of water quality by Nemerow pollution index method

According to Formula (2), the data of the three monitoring indicators in the Heilongtan Reservoir are calculated, and the Nemerow pollution index of each monitoring indicator in the Heilongtan Reservoir is shown in Figs 2–4 below. The water quality pollution index of Heilongtan Reservoir is between 0.36 and 0.51 in the past three years, and the water quality of the three monitoring sections of Dongfeng Canal, Longmiao and Sixin Village has not changed much. It can be seen that the water quality of Heilongtan Reservoir has always been classified as Category I, which is clean. From 2019 to 2021, the overall water quality of the reservoir has declined, but the change is not obvious.

**Table 5. Statistical results of various indicators in Heilongtan Reservoir in 2021.**

| Monitoring indicators | Monitoring section | | | |
|---|---|---|---|---|
| | The whole | | Longmiao | |
| | Average value | Measured category | Average value | Measured category |
| Permanganate Index | 2.48 | II | 2.48 | II |
| BOD$_5$ | 1.8 | I | 1.8 | I |
| Ammonia nitrogen | 0.202 | II | 0.202 | II |
| Petro | 0.005 | I | 0.005 | I |
| Volatile phenol | 0.00015 | I | 0.00015 | I |
| Hg | 0.00002 | I | 0.00002 | I |
| Pb | 0.002 | I | 0.002 | I |
| Cd | 0.0001 | I | 0.0001 | I |
| Anionic surfactant | 0.025 | I | 0.025 | I |
| Cr(hexavalent) | 0.002 | I | 0.002 | I |
| Fluoride(F⁻) | 0.199 | I | 0.199 | I |
| TP(P) | 0.015 | II | 0.015 | II |
| Cyanide | 0.002 | I | 0.002 | I |
| Sulfide | 0.0025 | I | 0.0025 | I |
| As | 0.0009 | I | 0.0009 | I |
| CODcr | 11.33 | I | 11.33 | I |
| Cu | 0.005 | I | 0.005 | I |
| Zn | 0.025 | I | 0.025 | I |
| Se(Tetravalent) | 0.0002 | I | 0.0002 | I |

Note: The unit of each index is mg/L.

**Table 6. Single factor index values for monitoring indicators (Pi).**

| Monitoring indicators | 2019 | | | | 2020 | | | | 2021 | |
|---|---|---|---|---|---|---|---|---|---|---|
| | Dongfeng Canal | Longmiao | Sixin Village | The whole | Dongfeng Canal | Longmiao | Sixin Village | The whole | Longmiao | The whole |
| Permanganate Index | 0.30 | 0.44 | 0.43 | 0.39 | 0.46 | 0.42 | 0.44 | 0.44 | 0.41 | 0.41 |
| BOD$_5$ | 0.4 | 0.4 | 0.4 | 0.38 | 0.40 | 0.43 | 0.43 | 0.43 | 0.45 | 0.45 |
| Ammonia nitrogen | 0.14 | 0.11 | 0.11 | 0.12 | 0.12 | 0.11 | 0.11 | 0.11 | 0.20 | 0.20 |
| Petro | 0.32 | 0.16 | 0.18 | 0.22 | 0.15 | 0.10 | 0.12 | 0.12 | 0.10 | 0.10 |
| Volatile phenol | 0.03 | 0.03 | 0.03 | 0.03 | 0.03 | 0.03 | 0.03 | 0.03 | 0.03 | 0.03 |
| Hg | 0.2 | 0.2 | 0.2 | 0.2 | 0.2 | 0.2 | 0.2 | 0.2 | 0.2 | 0.2 |
| Pb | 0.1 | 0.1 | 0.1 | 0.1 | 0.1 | 0.1 | 0.1 | 0.1 | 0.033 | 0.033 |
| Cd | 0.1 | 0.1 | 0.1 | 0.1 | 0.1 | 0.1 | 0.1 | 0.1 | 0.025 | 0.025 |
| Anionic surfactant | 0.144 | 0.125 | 0.125 | 0.131 | 0.125 | 0.125 | 0.125 | 0.125 | 0.125 | 0.125 |
| Cr(hexavalent) | 0.04 | 0.04 | 0.04 | 0.04 | 0.04 | 0.04 | 0.04 | 0.04 | 0.04 | 0.04 |
| Fluoride(F⁻) | 0.171 | 0.186 | 0.161 | 0.173 | 0.184 | 0.161 | 0.179 | 0.175 | 0.199 | 0.199 |
| TP(P) | 0.62 | 0.48 | 0.51 | 0.54 | 0.70 | 0.32 | 0.35 | 0.46 | 0.30 | 0.30 |
| Cyanide | 0.01 | 0.01 | 0.01 | 0.01 | 0.01 | 0.01 | 0.01 | 0.01 | 0.01 | 0.01 |
| Sulfide | 0.014 | 0.0125 | 0.0125 | 0.0128 | 0.0125 | 0.0125 | 0.0125 | 0.0125 | 0.0125 | 0.0125 |
| As | 0.021 | 0.022 | 0.020 | 0.021 | 0.025 | 0.024 | 0.025 | 0.024 | 0.017 | 0.017 |
| CODcr | 0.36 | 0.45 | 0.45 | 0.42 | 0.50 | 0.53 | 0.48 | 0.50 | 0.57 | 0.57 |
| Cu | 0.025 | 0.025 | 0.025 | 0.025 | 0.025 | 0.025 | 0.025 | 0.025 | 0.005 | 0.005 |
| Zn | 0.025 | 0.025 | 0.025 | 0.025 | 0.025 | 0.025 | 0.025 | 0.025 | 0.025 | 0.025 |
| Se(Tetravalent) | 0.02 | 0.02 | 0.02 | 0.02 | 0.02 | 0.02 | 0.02 | 0.02 | 0.02 | 0.02 |

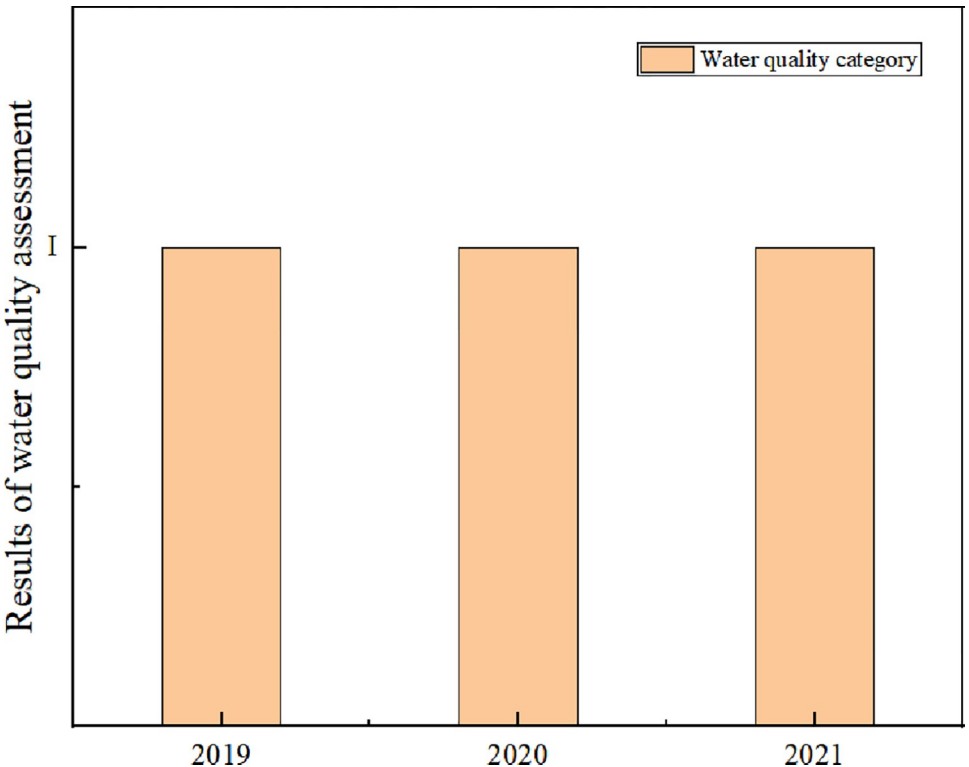

**Fig 1. Results of water quality assessment by single factor pollution index method.**

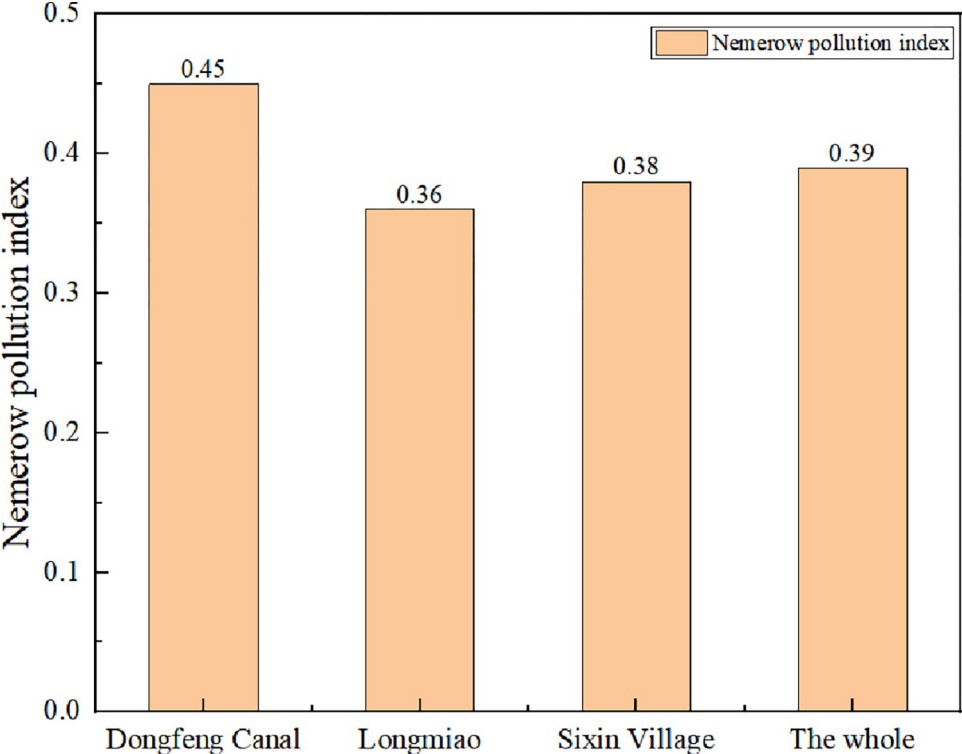

**Fig 2. Comparison of Nemerow pollution index of various monitoring sections of Heilongtan Reservoir in 2019.**

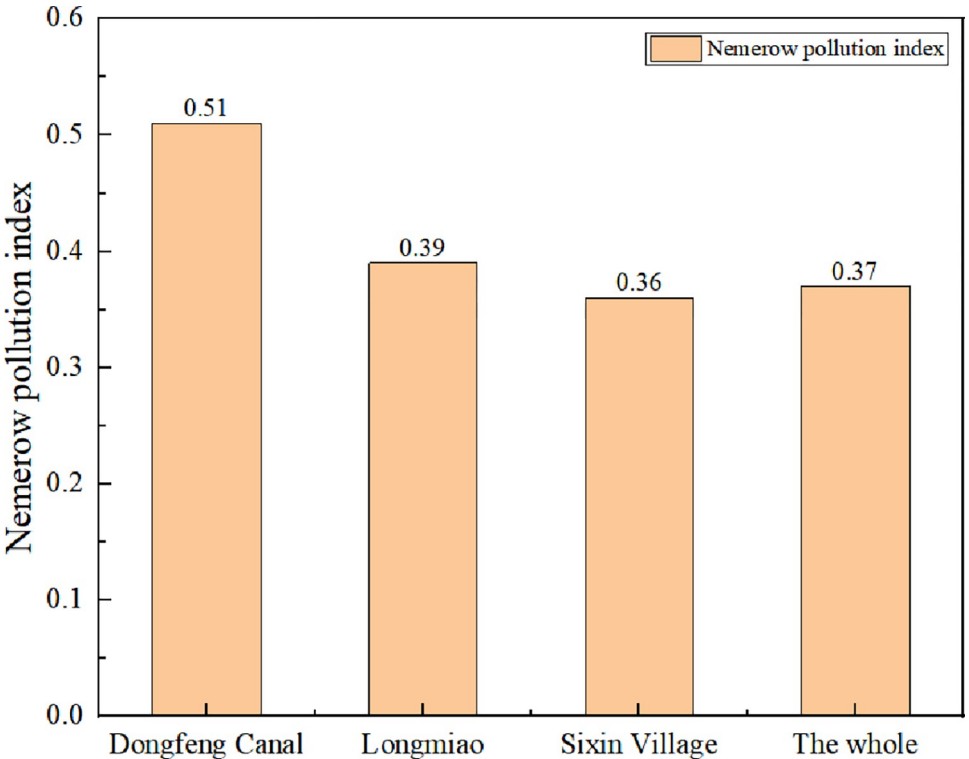

**Fig 3. Comparison of Nemerow pollution index of various monitoring sections of Heilongtan Reservoir in 2020.**

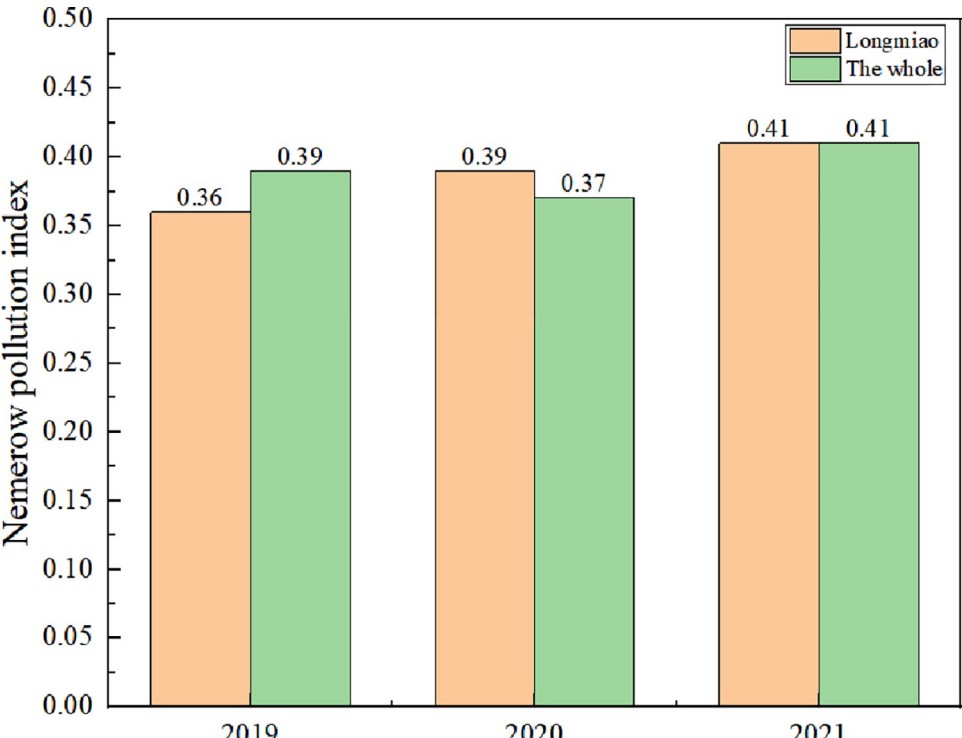

**Fig 4. Changes in Nemerow pollution index for the Longmiao and The whole from 2019 to 2021.**

**Table 7. Temporal and spatial changes of water quality in each monitoring section ofHeilongtan Reservoir from 2019 to 2020.**

| Monitoring section | $P_N$ | | T/% | S/% | | Datum section |
|---|---|---|---|---|---|---|
| | 2019 | 2020 | | 2019 | 2020 | |
| Dongfeng Canal | 0.45 | 0.51 | -0.13 | | | |
| Longmiao | 0.36 | 0.39 | -0.08 | 0.20 | 0.24 | Dongfeng Canal |
| Sixin Village | 0.38 | 0.36 | 0.05 | -0.06 | 0.08 | Longmiao |
| The whole | 0.39 | 0.37 | 0.05 | | | |

Compared to the evaluation results of the single factor pollution index method, the Nemerow index method is more comprehensive than the single factor index method for assessing water quality because it takes into the more polluting indicators with the average of the single factor indices, which accurately reflects the level of water pollution [20].

## Analysis of temporal and spatial variation characteristics of water quality

According to Eqs (3) and (4), the spatial and temporal rates of change in water quality were calculated for Dongfeng Canal, Longmiao and Sixin Village. The numerical magnitude of the temporal rate of change represents the degree of water quality change, and its positive or negative represents the good or bad water quality change. Tables 7 and 8 show that from 2019 to 2020, the water quality of the Dongfeng Canal deteriorates slightly (10%<T<20%), while the water quality of Longmiao, Sixin Village and the overall water quality remains largely unchanged (T<10%). There is also a slight deterioration in the overall water quality of Longmiao in 2021 compared to 2020 (10% < T<20%). In both 2019 and 2020, the water quality of Longmiao is significantly better than that of Dongfeng Canal (S>20%), and the water quality of Longmiao and Sixin Village does not change significantly (S<10%). In this study, because of the small number of monitoring points in 2021, individual monitoring points have a greater impact on the overall average water quality, making the overall water quality more variable, so more water quality monitoring sections should be set up in future studies to reduce the impact of chance factors on water quality assessment.

## Conclusion

The overall water quality of the Heilongtan Reservoir is great, and the water quality has not changed much in the past three years. During the three years 2019 to 2021, the monitoring indicators of the monitoring sections have reached the Category III water quality standard and above in the "Surface Water Environmental Quality Standard". According to the results of the evaluation of water quality using the single factor pollution index, the water quality of Heilongtan Reservoir in 2019, 2020 and 2021 is of Category I standard and belongs to no pollution. The results of the Nemerow pollution index method show that the water quality pollution

**Table 8. Temporal and spatial changes of water quality in each monitoring section ofHeilongtan Reservoir from 2020 to 2021.**

| Monitoring section | $P_N$ | | T/% |
|---|---|---|---|
| | 2020 | 2021 | |
| Dongfeng Canal | 0.51 | | |
| Longmiao | 0.39 | 0.41 | -0.05 |
| Sixin Village | 0.36 | | |
| The whole | 0.37 | 0.41 | -0.11 |

index for Heilongtan Reservoir during that time varied from 0.36 to 0.51, and the three monitoring sections in Dongfeng Canal, Longmiao, and Sixin Village had no appreciable changes in their water quality overall and all belonged to Category I water quality. From the perspective of time changes, from 2019 to 2020, only the water quality of the Dongfeng Canal monitoring section has slightly deteriorated (10%<T<20%), while the water quality of the rest of the monitoring sections is basically unchanged(T<10%). By 2021, the whole quality of the Heilongtan Reservoir has fluctuated, with a slight deterioration (10%<T<20%). From the perspective of spatial changes, in both 2019 and 2020, Longmiao's water quality is much superior to Dongfeng Canal's (S>20%), although there is no distinction between Longmiao and Sixin Village's water quality (S<10%), over these two years.

The three-year monitoring data of Heilongtan Reservoir were analysed by the single factor index method and the Nemerow pollution index method, so as to grasp the changes of water quality in Heilongtan Reservoir, with a view to providing a scientific basis for water protection in the reservoir. In the single factor index method, an important factor affecting the category of water quality evaluation is the selection of environmental control standard values. In the subsequent study, it is worth considering whether the control standard can be improved so as to obtain more stringent water quality evaluation results.

## Acknowledgments

We thank the editors and the reviewers for their useful feedback that improved this paper.

## Author Contributions

**Conceptualization:** Kai Su.

**Data curation:** Yingwei Xi, Guizhi Li.

**Methodology:** Kai Su.

**Project administration:** Kai Su.

**Writing – original draft:** Qin Wang, Linxiao Li, Rong Cao.

**Writing – review & editing:** Kai Su.

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
