## [Decision Letter · Decision Letter 0]

18 Jul 2022

PONE-D-22-12469Water Quality Assessment Based on Nemerow Pollution Index Method: A Case Study of Heilongtan Reservoir in Central Sichuan Province, ChinaPLOS ONE

Dear Dr. Su,

Thank you for submitting your manuscript to PLOS ONE. After careful consideration, we feel that it has merit but does not fully meet PLOS ONE’s publication criteria as it currently stands. Therefore, we invite you to submit a revised version of the manuscript that addresses the points raised during the review process.

We look forward to receiving your revised manuscript.

Kind regards,

Xiaoshan Zhu, Ph.D.

Academic Editor

PLOS ONE

Journal Requirements:

“This study is supported by Sichuan Science and Technology Program (2021YFS0284).”

“This study is supported by Sichuan Science and Technology Program (2021YFS0284).”

“This study is supported by Sichuan Science and Technology Program (2021YFS0284).”

Reviewers' comments:

Reviewer's Responses to Questions

**Comments to the Author**

1. Is the manuscript technically sound, and do the data support the conclusions?

Reviewer #1: Yes

Reviewer #2: Yes

2. Has the statistical analysis been performed appropriately and rigorously? 

Reviewer #1: Yes

Reviewer #2: Yes

3. Have the authors made all data underlying the findings in their manuscript fully available?

Reviewer #1: Yes

Reviewer #2: Yes

4. Is the manuscript presented in an intelligible fashion and written in standard English?

Reviewer #1: Yes

Reviewer #2: No

5. Review Comments to the Author

Reviewer #1: This paper uses the water quality monitoring data of Heilongtan Reservoir from 2019 to 2021, adopts the single factor pollution index method and Nemerow pollution index method to evaluate and analyze the water quality changes of Heilongtan Reservoir in three years, and explores the impact of human activities on the water quality of Heilongtan ReservoirIn addition, the article is easy to understand and has a clear framework system. However, there are some detailed mistakes in the article I have displayed as follows. I suggest that this paper can be better after some modifications.

Specific comments:

（1）The first sentence of the first paragraph of the Introduction is the same as the last sentence, please enrich the content.

（2）The same phrase is used in multiple places in the text, please polish the text.

（3）In line 62, just use an example to prove the point is not enough.

（4）Line 96, please reduce the use of short sentences and merge short sentences to express them more professionally.

（5）Line 108, about “Nemerrow pollution index method”, needs more introduction.

（6）The width of the column chart rectangle in the article is too wide, please adjust it to make it look more beautiful.

（7）Please check the presentation of the article to avoid difficult to read questions.

（8）Please check the use of passive voice in the article. There are some misuses of passive voice.

（9）Many expressions in the article are Chinese expressions, please modify them to make the expressions more native.

（10）At the end of the paper, you can write the further meaning and some prospects of your research.

（11）The content of the article lacks innovation, please emphasize its innovation and research significance.

Reviewer #2: Accurate evaluation of water quality is the basis of realizing river and lake health goals. In order to achieve accurate evaluation of river and lake water environment, Nemerow Pollution Index Method was used to evaluate water quality. The research method of this paper can be used as a reference for water quality evaluation of rivers and lakes. Where is the innovation of the system? What is the use value of the research method? Furthermore, there are more details (not limited) that should be discussed as following.

1. In the abstract, add the research method and research conclusion.

2. We can’t get the conclusion of “Protection work is particularly important to the survival and development of human beings”.

3. Add the difference analysis among methods used for water environment quality evaluation.

4. Include the limitation of the study in Introduction.

5. In the section of 2.1, add the current status of water quality in the Heilongtan Reservoir.

6. I recommend giving the data in the form of a table.

7. The paper lacks content for discussion. The research process of the paper is detailed and the research results are fruitful, but we can not judge the correctness of the calculation results.

8. Add the discussion on the rationality of research results and the applicability of research methods.

6. PLOS authors have the option to publish the peer review history of their article (what does this mean?). If published, this will include your full peer review and any attached files.

Reviewer #1: No

Reviewer #2: No

---

## [Author Response · Author response to Decision Letter 0]

26 Jul 2022

Dear Reviewers:

We appreciate your comments and ideas on our manuscript, which we used to make significant revisions to the original. In the amended version, all changes have been highlighted in the revised manuscript by formatting the text with yellow color. We hope the reviewers are pleased with our responses to the "comments" and the revisions we made to the original manuscript.

---

## [Decision Letter · Decision Letter 1]

2 Aug 2022

PONE-D-22-12469R1Water Quality Assessment Based on Nemerow Pollution Index Method: A Case Study of Heilongtan Reservoir in Central Sichuan Province, ChinaPLOS ONE

Dear Dr. Su,

Thank you for submitting your manuscript to PLOS ONE. After careful consideration, we feel that it has merit but does not fully meet PLOS ONE’s publication criteria as it currently stands. Therefore, we invite you to submit a revised version of the manuscript that addresses the points raised during the review process.

We look forward to receiving your revised manuscript.

Kind regards,

Xiaoshan Zhu, Ph.D.

Academic Editor

PLOS ONE

Journal Requirements:

Reviewers' comments:

Reviewer's Responses to Questions

**Comments to the Author**

1. If the authors have adequately addressed your comments raised in a previous round of review and you feel that this manuscript is now acceptable for publication, you may indicate that here to bypass the “Comments to the Author” section, enter your conflict of interest statement in the “Confidential to Editor” section, and submit your "Accept" recommendation.

Reviewer #1: All comments have been addressed

Reviewer #2: All comments have been addressed

2. Is the manuscript technically sound, and do the data support the conclusions?

Reviewer #1: Yes

Reviewer #2: Yes

3. Has the statistical analysis been performed appropriately and rigorously? 

Reviewer #1: Yes

Reviewer #2: Yes

4. Have the authors made all data underlying the findings in their manuscript fully available?

Reviewer #1: Yes

Reviewer #2: Yes

5. Is the manuscript presented in an intelligible fashion and written in standard English?

Reviewer #1: Yes

Reviewer #2: Yes

6. Review Comments to the Author

Reviewer #1: I think the author has completed the most comments, However the structure of this paper is still need some work. I think there are too many tables in this paper. Can you try to draw more figures for this paper.

Reviewer #2: Accurate evaluation of water quality is the basis of realizing river and lake health goals. In order to achieve accurate evaluation of river and lake water environment, Nemerow Pollution Index Method was used to evaluate water quality. The research method of this paper can be used as a reference for water quality evaluation of rivers and lakes.

7. PLOS authors have the option to publish the peer review history of their article (what does this mean?). If published, this will include your full peer review and any attached files.

Reviewer #1: No

Reviewer #2: No

---

## [Author Response · Author response to Decision Letter 1]

5 Aug 2022

Dear Editor and Reviewers:

We appreciate your comments and ideas on our manuscript, which we used to make significant revisions to the original. In the amended version, all changes have been highlighted in the revised manuscript by formatting the text with yellow color. We hope the reviewers are pleased with our responses to the "comments" and the revisions we made to the original manuscript.

---

## [Editor Report · Decision Letter 2]

8 Aug 2022

Water Quality Assessment Based on Nemerow Pollution Index Method: A Case Study of Heilongtan Reservoir in Central Sichuan Province, China

PONE-D-22-12469R2

Dear Dr. Su,

We’re pleased to inform you that your manuscript has been judged scientifically suitable for publication and will be formally accepted for publication once it meets all outstanding technical requirements.

Kind regards,

Xiaoshan Zhu, Ph.D.

Academic Editor

PLOS ONE
---

## [Editor Report · Acceptance letter]

10 Aug 2022

PONE-D-22-12469R2 

Water quality assessment based on Nemerow pollution index method: A case study of Heilongtan reservoir in central Sichuan province, China 

Dear Dr. Su:

I'm pleased to inform you that your manuscript has been deemed suitable for publication in PLOS ONE. Congratulations! Your manuscript is now with our production department. 

Kind regards, 

on behalf of

Dr. Xiaoshan Zhu 

Academic Editor

PLOS ONE